# SSR: Enhancing Depth Perception in Vision-Language Models via Rationale-Guided Spatial Reasoning

**Yang Liu**[1,†], **Ming Ma**[3,†], **Xiaomin Yu**[4,†], **Pengxiang Ding**[1,2,†,§],
**Han Zhao**[1,2], **Mingyang Sun**[1,2,5], **Siteng Huang**[2], **Donglin Wang**[1*]
[1]Westlake University, [2]Zhejiang University, [3]Harbin Institute of Technology,
[4]The Hong Kong University of Science and Technology (Guangzhou),
[5]Shanghai Innovation Institute
{liuyang67, wangdonglin}@westlake.edu.cn

## Abstract

Despite impressive advancements in Visual-Language Models (VLMs) for multi-modal tasks, their reliance on RGB inputs limits precise spatial understanding. Existing methods for integrating spatial cues, such as point clouds or depth, either require specialized sensors or fail to effectively exploit depth information for higher-order reasoning. To this end, we propose a novel Spatial Sense and Reasoning method, dubbed SSR, a novel framework that transforms raw depth data into structured, interpretable textual rationales. These textual rationales serve as meaningful intermediate representations to significantly enhance spatial reasoning capabilities. Additionally, we leverage knowledge distillation to compress the generated rationales into compact latent embeddings, which facilitate resource-efficient and plug-and-play integration into existing VLMs without retraining. To enable comprehensive evaluation, we introduce a new dataset named SSR-COT, a million-scale visual-language reasoning dataset enriched with intermediate spatial reasoning annotations, and present SSRBENCH, a comprehensive multi-task benchmark. Extensive experiments on multiple benchmarks demonstrate SSR substantially improves depth utilization and enhances spatial reasoning, thereby advancing VLMs toward more human-like multi-modal understanding. Project page: https://yliu-cs.github.io/SSR.

## 1 Introduction

VLMs represent a pivotal advancement in bridging the gap between image and natural language, demonstrating astounding capabilities across myriad multi-modal tasks [1–7]. Nevertheless, relying solely on RGB is inadequate for accurately capturing spatial information such as relative positions and distances, which presents inherent limitations in capturing precise spatial relationships, thereby constraining the capacity of VLMs to comprehend complex scenes. Consequently, enhancing the ability of VLMs to understand and reason about spatial relationships is essential for critical real-world applications, particularly in robotics.

Recent advancements in VLMs have catalyzed research on explicitly incorporating spatial information to enhance model performance. While some methods leverage point cloud data for improved spatial understanding [8–10], they typically rely on specialized sensors (e.g., LiDAR) that are impractical in scenarios restricted to monocular RGB images. In this context, monocular depth estimation has emerged as a compelling alternative, particularly with the proliferation of generative methods [11, 12]. These methods enable the acquisition of high-quality depth images from standard 2D images through

---

[*]Corresponding author. [†]Equal contribution. [§]Project lead.

39th Conference on Neural Information Processing Systems (NeurIPS 2025).

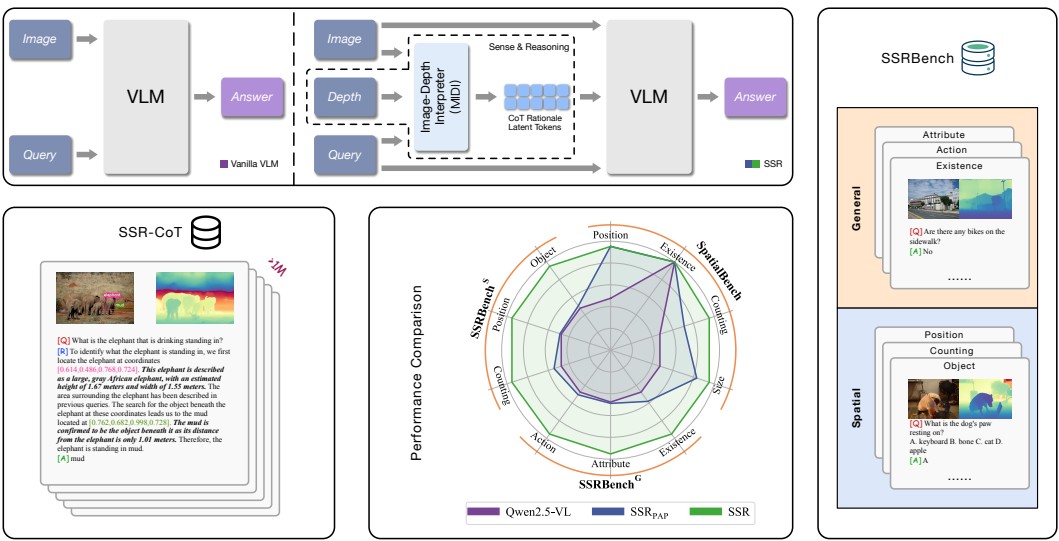

Figure 1: Unlike conventional VLMs, SSR integrates depth perception to enhance spatial reasoning. We introduce a curated dataset SSR-COT and benchmark SSRBENCH, demonstrating significant improvements in spatial reasoning tasks.

various pre-trained models [13–16], eliminating additional hardware requirements. By leveraging visual encoders pretrained on RGB images, depth features can be efficiently encoded and seamlessly integrated into VLMs, offering a promising pathway for enhancing spatial awareness.

However, a critical limitation of current methods lies in their superficial utilization of depth information [8–10, 17–20]. Unlike humans, who intuitively employ depth as an integral component within broader reasoning processes, existing methods incorporate depth explicitly without capitalizing on its inferential value [17]. Consider a query such as *Are objects A and B far apart?* Human cognition naturally analyzes the spatial relationship between objects and then leverages this understanding to inform subsequent reasoning. This implicit reasoning process underscores the necessity for more sophisticated integration of depth information into VLMs, not merely as supplementary input, but as a fundamental component that facilitates complex spatial reasoning. Developing methodologies that emulate this human-like implicit utilization of depth could substantially enhance VLM capabilities.

To this end, we propose SSR, a novel paradigm designed to redefine the integration of depth information within VLMs. Specifically, SSR translates raw depth data into a structured *rationale language*, providing an interpretable intermediate representation that bridges low-level depth perception and higher-level reasoning. This rationale-based language facilitates VLMs in generating outputs that are both more accurate and contextually appropriate, while also enabling the previously underutilized inferential capabilities inherent to spatial depth information. By converting modality-specific depth data into semantically rich and inherently aligned representations, SSR effectively overcomes the interpretability limitations associated with traditional approaches. Consequently, this method significantly enhances the utilization of depth information, laying the groundwork for more robust and human-like spatial reasoning capabilities within contemporary VLMs. To further enhance the efficiency of rationale-language utilization, we transform depth information into a compact latent embedding. Specifically, we apply a knowledge-distillation strategy to compress rationale-language representations into concise latent embeddings [21]. Dissimilar to vanilla Chain-of-Thought (CoT) methods [22, 21, 23, 24] that rely primarily on textual explanations, our distillation strategy significantly reduces computational overhead while preserving the depth and inferential richness inherent to rationale-based representations [25–27]. Importantly, this module can be seamlessly integrated into existing VLMs via a training-free mechanism, highlighting the flexibility and broad applicability of the proposed framework. To achieve SSR, we first curate SSR-COT, a million-level vision-language spatial reasoning dataset that facilitates depth-aware reasoning and provides a robust foundation for developing sophisticated spatial reasoning models. To validate our approach, we perform extensive experiments across multiple benchmarks. Specifically, we also evaluate on our benchmark SSR-BENCH, which comprises six distinct tasks spanning both general and spatial domains. Extensive

experiments and analysis demonstrate that SSR substantially enhances spatial reasoning capabilities across diverse tasks, highlighting the effectiveness and broad utility of our proposed method.

Overall, our principal contributions in this paper are illustrated in Figure 1 and summarized as follows:
• We propose an efficient VLM, dubbed SSR, capable of simultaneously performing depth perception and spatial reasoning, and generating answers based on implicit reasoning rationales.
• We introduce SSR-COT, a million-scale visual-language reasoning dataset enriched with intermediate spatial reasoning annotations, and present SSRBENCH, a comprehensive multi-task benchmark.
• Extensive experiments and solid analysis across various benchmarks demonstrate our SSR can efficiently and dramatically enhance the spatial understanding of existing VLMs.

## 2 Related Work

### 2.1 Visual-Language Models

LLMs [24, 28–38] have led to major advancements in Natural Language Processing (NLP) tasks, and also have incited interest in developing VLMs. Building a unified LLM with visual inputs for visual language tasks thus remains one of the most important desiderata for VLMs. Over the last few years, VLMs achieved significant performance improvements in multi-modal tasks by integrating a pre-trained visual encoder and projecting the feature into semantic space into LLMs as well as training on large-scale multi-modal question-answering pairs [1, 2, 39–43]. This straightforward method can work well for general tasks, but expecting the model to deduce answers for more complex tasks without deep reasoning can be daunting.

### 2.2 Multi-Modal Reasoning

LLMs display an emergent capability for step-by-step reasoning through in-context learning, a phenomenon referred to as CoT reasoning. Such reasoning significantly enhances the performance of LLMs on complex reasoning tasks [22, 44, 45]. Concurrently, notable advancements have also occurred in multi-modal CoT research, a paradigm appealing due to its similarity to human problem-solving behaviors [46, 47]. Current research efforts in multi-modal CoT primarily emphasize the construction of intermediate reasoning rationale datasets to train image-text reasoning models. Many existing studies adopt rich textual captions and detailed descriptions as intermediate rationales [48–51]. Beyond text-based rationales, recent approaches have leveraged multi-modal rationales for more comprehensive reasoning [52–56]. However, existing multi-modal CoT methods primarily focus on tasks involving code generation, mathematical problem solving, and general question answering, which require sophisticated reasoning to achieve accurate responses. In contrast, this paper introduces an efficient CoT method that leverages depth images to enhance the performance of VLMs, particularly by improving spatial understanding.

### 2.3 Spatial Intelligence

Spatial reasoning is an essential capability for VLMs and has therefore been included in several Visual Question Answering (VQA) benchmarks [57–60]. However, the majority of existing VLMs [7, 61–64] are primarily trained on two-dimensional images paired with textual data, a setting that inherently lacks comprehensive spatial information. Consequently, these models exhibit limited performance in spatial reasoning tasks. To overcome this limitation, recent works such as SpatialVLM [10], SpatialRGPT [65] and RoboRefer [66] have sought to improve the spatial reasoning capacity of VLMs by compiling specialized spatially-oriented question-answer datasets and fine-tuning models accordingly. Nevertheless, despite these advancements, prior approaches largely neglect the integration of language-based reasoning capabilities within the spatial reasoning framework. This omission hampers the effectiveness of existing VLMs in addressing more complex tasks, particularly those requiring intricate or multi-step reasoning processes.

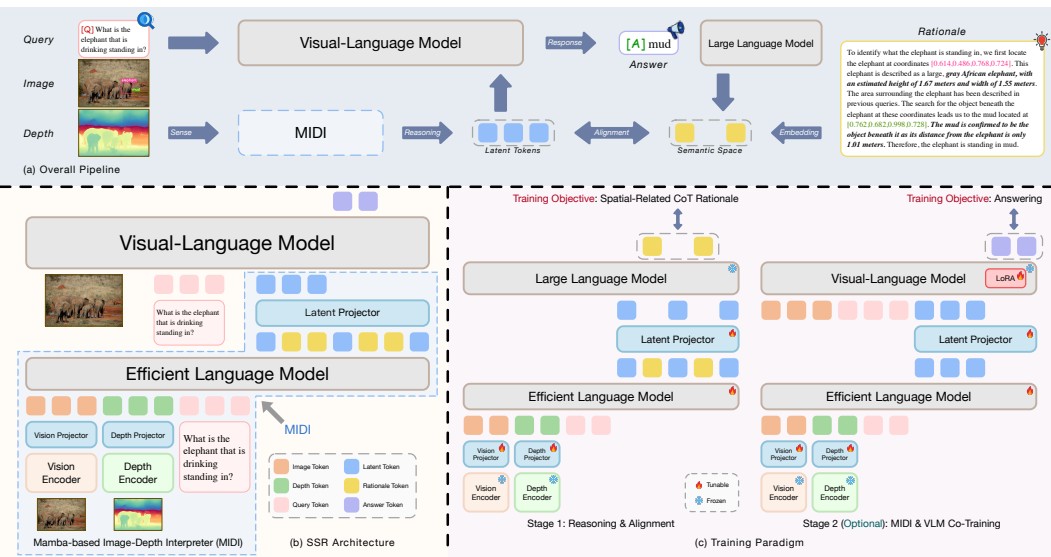

Figure 2: Schematic of SSR framework. (a) Overall pipeline. (b) Full architecture of SSR, comprising the MIDI module followed by the VLM. (c) Two training stages of the SSR. In the stage 1, the LLM provides alignment supervision for the MIDI module, whereas the stage 2 is optional.

## 3 Methodology

### 3.1 Architecture

The primary goal of our proposed SSR is to effectively leverage the reasoning capability of efficient language models effectively enhance the depth understanding and spatial reasoning capability for existing VLMs. The overall framework is illustrated in Figure 2.

#### 3.1.1 Image-Depth Interpreter

To achieve comprehensive spatial understanding via depth interpretation, we propose a simple yet effective plug-and-play module entitled Mamba-based Image-Depth Interpreter (MIDI). MIDI generates enriched depth-aware latent token representations, providing essential spatial reasoning information before feeding these tokens into the VLM.

Given an input image $X_V \in \mathbb{R}^{H \times W \times 3}$ and a corresponding textual query $X_T$, we first utilize a pretrained monocular depth estimation model, Depth Pro [15], to produce a depth $X_D \in \mathbb{R}^{H \times W \times 1}$ in the image. Subsequently, image features $Z_V$ and depth features $Z_D$ are extracted from $X_V$ and $X_D$, respectively. Specifically, we employ pre-trained CLIP ViT-L/14 [67, 68] as the visual encoder $\mathcal{E}_V$, and SigLIP [69] as the depth encoder $\mathcal{E}_D$: $H_\alpha = \mathcal{E}_\alpha(X_\alpha), \alpha \in \{V, D\}$. Then we apply Multi-Layer Perceptron (MLP) modules, comprising two fully connected layers with GELU [70] activation, as projectors $\phi_V$ and $\phi_D$, transforming these visual features into the semantic embedding space compatible with the subsequent efficient language model: $Z_\alpha = \phi_\alpha(H_\alpha), \alpha \in \{V, D\}$. To jointly encode visual and depth information conditioned on the textual query, we introduce an intermediate reasoning module implemented as the Mamba-based language model [71], denoted as $f_{\mathrm{LM}}$. This module produces latent tokens representing intermediate spatial rationales: $H_R = f_{\mathrm{LM}}(Z_V, Z_D, X_T)$. Specifically, we uniformly insert several additional special tokens into the rationales to facilitate the knowledge distillation process and encode textual tokens into latent representations [21]. Finally, similar to previous steps, we apply an additional latent projection module $\phi_R$ to map these latent rationale tokens into another semantic embedding space, matching the dimensionality of the word embeddings used in the subsequent VLM: $Z_R = \phi_R(H_R)$.

Hence, our proposed MIDI module generates a sequence of spatial-aware latent tokens $Z_R$. These tokens can easily be plugged into the query sequence for existing VLMs, effectively injecting depth-based spatial reasoning information and significantly enhancing the spatial understanding capabilities.

Table 1: The mixture detail of SSR-CoT dataset. SSR-CoT consist over 1 million image-depth-question-rationale-answer pairs, where the rationale containing rich spatial-related knowledge the enhance Visual-Language Models (VLMs).

| Dataset | Source | Size | Dataset | Source | Size |
|---|---|---|---|---|---|
| LLaVA-CoT [51] | ShareGPT4V [72] | 31.3k | Visual-CoT [52] | Flickr30k [73] | 136k |
| | ChartQA [74] | 17.2k | | GQA [57] | 88k |
| | A-OKVQA [75] | 16.1k | | Visual7W [76] | 43k |
| | AI2D [77] | 11.4k | | OpenImages [78] | 43k |
| | GeoQA+ [79] | 11.4k | | Birds-200-2021 [80] | 10k |
| | ScienceQA [81] | 5.6k | | VSR [82] | 3k |
| | DocVQA [83] | 4.0k | | SCREENED TOTAL | *289k* |
| | PISC [84] | 1.0k | VoCoT [53] | GQA [57] | 72k |
| | CLEVR [85] | 0.5k | | LLaVA-Instruct [86] | 6k |
| | CLEVR-Math [87] | 0.5k | | LVIS [88] | 2k |
| | TOTAL | *98k* | SCREENED ONE-TURN TOTAL | | *317k* |
| SpatialQA [17] | Bunny [89] | 695k | | OXE [90] | 7.5k |
| | | | | SCREENED TOTAL | *501k* |

### 3.1.2 Spatial Sense and Reasoning

Our proposed MIDI module fully leverages spatial information derived from the depth images and generates a latent representation $Z_R$, which encodes intermediate reasoning rationales essential for producing the response. Subsequently, we input these latent tokens $Z_R$, alongside the original image $X_V$ and textual question $X_T$, into an existing VLM $f_{\text{VLM}}$ to generate the answer $Y_A$ in an auto-regressive manner: $Y_A = f_{\text{VLM}}(X_V, Z_R, X_T)$.

## 3.2 Training Paradigm

For training the proposed SSR, we adopt a two-stage procedure, as illustrated on the bottom-right side of Figure 2. In Stage 1, we train the underlying MIDI module to generate rationale latent tokens and project them into the language semantic space. In Stage 2, we conduct joint training of the MIDI module and existing Vision-Language Models (VLMs) to further enhance performance. Notably, Stage 2 is optional due to the modular and plug-and-play nature of our MIDI module, enabling straightforward integration into existing VLM frameworks.

### 3.2.1 Stage 1: Reasoning and Alignment

At the initial stage, we aim to train an efficient language model within the MIDI module, enabling it to generate and encode coherent thought processes represented by a sequence of features consistently aligned with the natural language semantic space. To this end, each training sample at this phase includes a detailed and accurate rationale $Y_R$ as the ground truth. After feeding latent tokens produced by the MIDI module into the subsequent LLM, we require the LLM to reconstruct the original textual rationale solely from these latent representations. Training the LLM for precise rationale recovery depends not only on accurate reasoning capabilities of the MIDI module itself, but also on successfully projecting latent tokens into a semantic space consistent with the frozen-state LLM.

The learning objective for Stage 1 is defined by the standard causal modeling loss, given by:

$$\mathcal{L}_1(\theta) = -\mathbb{E}_{(X_V, X_D, X_T, Z_R, Y_R) \sim D} \left[ \frac{1}{|Y_R|} \sum_{i=1}^{|Y_R|} \log P_\theta(Y_{R,i} \mid X_V, X_D, X_T, Z_R, Y_{R,<i}) \right]. \tag{1}$$

Following this training stage, latent tokens $Z_R$ generated by the MIDI module can be readily integrated into existing VLM image-text sequences, thereby enhancing their spatial understanding capabilities.

### 3.2.2 Stage 2: Co-Training

To further enhance the performance of SSR, we jointly train the MIDI module along with existing VLMs. Similar to instruction-tuning, we discard intermediate rationales in the second training stage and allow the VLM to directly generate the final answer. In this setting, accurate answer generation by the VLM requires not only effective reasoning from the MIDI module but also the capacity of

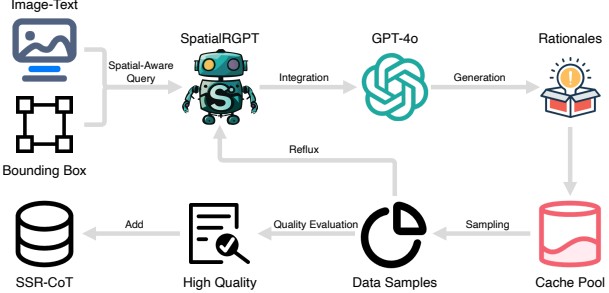

Figure 3: Schematic of SSR-CoT annotation pipeline.

Table 2: Quality evaluation for SSR-CoT dataset. We conduct the evaluation based on the powerful visual-language model Qwen2.5-VL-7B-Instruct [91].

| Rationale | Accuracy | Score |
|:---:|:---:|:---:|
| ✗ | 67.80 | 3.6721 |
| ✓ | 79.42 (↑11.62) | 4.1289 (↑0.4568) |

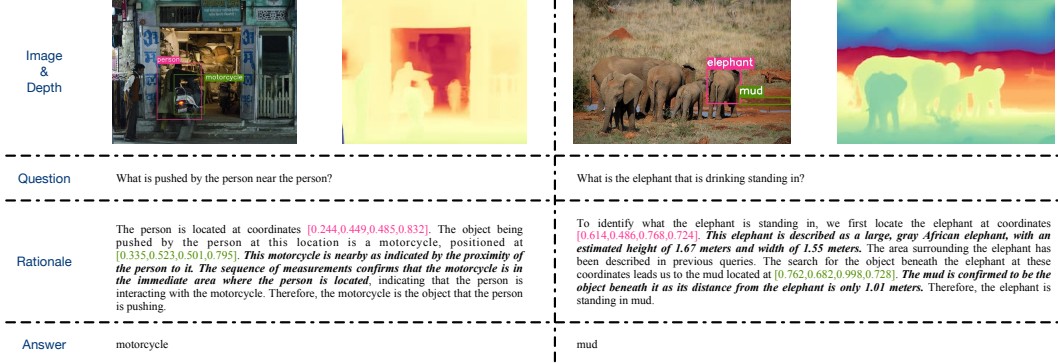

Figure 4: Illustrative samples of SSR-CoT dataset.

VLM to comprehend and utilize the reasoning information. Specifically, the second-stage learning objective is formulated as a standard cross-entropy loss for auto-regressive generation of the final answer $Y_A$:

$$\mathcal{L}_2(\theta) = -\mathbb{E}_{(X_V, X_D, X_T, Y_A) \sim D} \left[ \frac{1}{|Y_A|} \sum_{j=1}^{|Y_A|} \log P_\theta(Y_{A,j} \mid X_V, X_D, X_T, Y_{A,<j}) \right]. \quad (2)$$

Since the rationale serving as the ground truth for supervised learning is omitted during Stage 2 training, we can incorporate additional VQA pairs to expand the dataset, thereby enhancing the generalization capability of the model. Furthermore, training during this stage is optional due to the modular plug-and-play nature of the MIDI module.

## 4 Experimentation

### 4.1 SSR-CoT Collection

There is a scarcity of visual-language CoT datasets with detailed reasoning processes annotations to train the SSR model for depth perception and spatial understanding. Therefore, we curate a new dataset from existing VQA datasets, resulting in over a total of 1 million image-depth-question-rationale-answer pairs. There are four dataset sources we integrated: (1) **LLaVA-CoT** [51]: Systematic and structured reasoning visual-language CoT dataset, including general and science-targeted VQA data source. (2) **Visual-CoT** [52]: Multimodal CoT dataset that takes the bounding box as an intermediate thinking step, including general, relation reasoning and fine-grained science-targeted VQA data source. (3) **VoCoT** [53]: Fine-grained image-text CoT dataset that rationale provides detailed relationships between various objects with bounding box, including general and relation reasoning VQA data source. (4) **SpatialQA** [17]: Spatial QA dataset for sufficient utilization, including depth-related and robotic-related VQA data sources.

To generate visual-language reasoning data enriched with spatial information, we follow a multi-step process, as shown in Figure 3. First, we extract depth estimations from raw images using Depth Pro [15]. For the LLaVA-CoT [51] source, this is the only preprocessing step performed. Second, for

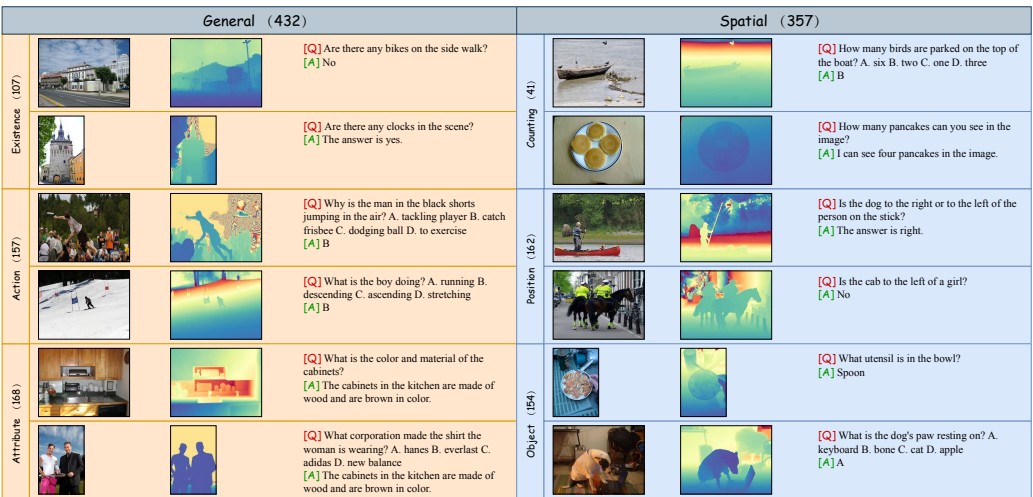

Figure 5: Examples for each task within the benchmark SSRBENCH.

datasets such as VoCoT [53] and SpatialQA [17], we refine long-form conversations by extracting concise, one-turn question-answer pairs. Third, we leverage SpatialRGPT [65] to comprehensively mine precise spatial attributes within images, such as object size, distance, and relative positioning, based on intermediate reasoning steps, including bounding box annotations from Visual-CoT [52] and VoCoT [53]. Finally, we employ GPT-4o [92] to integrate all extracted information, generating detailed reasoning processes that enhance spatial understanding. Notably, we also incorporate cache pools and perform sampling quality checks within iterative loops to ensure the high quality of the generated data. Specifically, similar to the quality-assessment protocol shown in Table 1, we randomly draw 10% of the cached samples and evaluate VQA accuracy both with and without their generated rationales. Rationales that improve accuracy are retained and incorporated into the final SSR-CoT dataset; those that degrade performance are discarded, and the all samples in cache are re-submitted for re-annotation. Overall, we compile approximately 1.2 million preprocessed data samples into SSR-CoT dataset. Figure 4 illustrates several data samples from the SSR-CoT dataset. Each data instance within SSR-CoT comprises the original image, an associated question-answer pair, the corresponding estimated depth information, and a rationale. The rationale incorporates fundamental reasoning steps used in question-answering tasks and provides detailed spatial reasoning to support accurate answer generation.

To evaluate the quality of SSR-CoT, we conducted an assessment based on the performance of the Qwen2.5-VL-7B-Instruct [91] on the VQA task. This evaluation was carried out on a randomly selected subset comprising approximately 1% of the full dataset, corresponding roughly to 10k samples. Performance metrics include accuracy as well as a quantitative score ranging from 0 to 5, both are produced using the LLM-Assistant powered by the Qwen2.5-14B-Instruct-1M [38, 93]. Further methodological details regarding the evaluation process are described in Appendix D. As presented in Table 2, responses generated with intermediate rationales demonstrate an accuracy improvement of more than 10% compared to direct question-answering. This finding indicates that the intermediate reasoning rationales annotated in our dataset are of high quality and effectively enhance the question-answering performance of VLMs.

## 4.2 SSRBENCH Construction

Currently, there are no established benchmarks specifically designed for evaluating spatial understanding and reasoning capabilities on image-text pairs. To address this gap, we propose SSRBENCH, a novel evaluation benchmark created from the SSR-CoT dataset. Importantly, the data incorporated into SSRBENCH will be fully removed from SSR-CoT to prevent overlap. SSRBENCH consists of two primary categories, general understanding and spatial understanding, allowing simultaneous evaluation of VLM performance in both general question answering and spatial reasoning tasks. Each category contains three distinct evaluation tasks, with detailed sample sizes provided in Appendix E.

Table 3: Performance comparison on SpatialBench [17] and our SSRBench. SSRBench G and SSRBench S denote general and spatial tasks, respectively.

| Method | Size | | SpatialBench [17] | | | | SSRBENCH G | | | SSRBENCH S | | |
|---|---|---|---|---|---|---|---|---|---|---|---|---|
| | Mamba | VLM | Position | Existence | Counting | Size | Existence | Attribute | Action | Counting | Position | Object |
| PROPRIETARY | | | | | | | | | | | | |
| GPT-4o-mini [92] | N/A | N/A | 47.1 | 75.0 | 70.5 | 21.7 | 72.0 | 48.2 | 63.1 | 53.7 | 44.4 | 46.1 |
| Claude-3.5-Haiku [99] | N/A | N/A | 55.9 | 65.0 | 72.2 | 26.7 | 51.4 | 52.2 | 43.7 | 42.0 | 34.1 | 38.3 |
| OPEN-SOURCE | | | | | | | | | | | | |
| SpatialVLM [10, 100] | N/A | 3B | 52.9 | 80.0 | 77.1 | 28.3 | 31.7 | 58.3 | 63.7 | 31.7 | 55.8 | 65.4 |
| LLaVA-1.5 [2] | N/A | 7B | 44.1 | 45.0 | 82.8 | 30.0 | 81.3 | 64.3 | 66.9 | 43.9 | 63.6 | 63.6 |
| LLaVA-NeXT [86] | N/A | 7B | 47.1 | 75.0 | 84.0 | 20.0 | 83.2 | 66.7 | 69.4 | 51.2 | 69.8 | 64.9 |
| LLaVA-NeXT [86] | N/A | 13B | 47.1 | 75.0 | 82.9 | 20.0 | 86.9 | 69.6 | 71.3 | 41.5 | 69.8 | 53.2 |
| SpatialBot [17] | N/A | 3B | 50.0 | 80.0 | 86.7 | 25.0 | 75.7 | 61.3 | 67.5 | 39.0 | 74.1 | 61.7 |
| Emu3 [101] | N/A | 8B | 47.1 | 20.0 | 10.0 | 25.0 | 58.9 | 35.7 | 37.6 | 19.5 | 51.9 | 37.0 |
| Qwen2.5-VL [91] | N/A | 3B | 55.9 | 80.0 | 76.4 | 25.0 | 66.4 | 58.9 | 63.1 | 34.1 | 60.5 | 51.9 |
| Qwen2.5-VL [91] | N/A | 7B | 61.8 | 80.0 | 87.1 | 30.0 | 75.7 | 62.5 | 70.1 | 43.9 | 61.7 | 55.2 |
| **SSR (Ours)** | 130M | 3B | **64.7** | 80.0 | 82.9 | **31.7** | 83.2 | **82.1** | 72.6 | 51.2 | 83.3 | 74.7 |
| **SSR (Ours)** | 130M | 7B | **64.7** | **85.0** | **90.2** | 28.3 | **90.7** | 79.2 | **76.4** | **65.9** | **84.6** | **77.9** |

Table 4: Performance improvement of SSR compared to the backbone model. SpatialBench [17], SSRBENCH, and CV-Bench [102] report average.

| Method | Size | | SpatialBench [17] | SSRBENCH | | CV-Bench [102] | VSR [82] | | What's Up [103] |
|---|---|---|---|---|---|---|---|---|---|
| | Mamba | VLM | | General | Spatial | | Random | Zero-Shot | |
| Qwen2.5-VL [91] | N/A | 3B | 59.3 | 62.8 | 48.8 | 67.0 | 73.0 | 76.4 | 85.4 |
| **SSR (Ours)** | 130M | 3B | 64.8 (↑5.4) | 79.3 (↑16.5) | 69.7 (↑20.9) | 68.9 (↑1.9) | 78.6 (↑5.6) | 82.9 (↑6.5) | 87.9 (↑2.5) |
| Qwen2.5-VL [91] | N/A | 7B | 64.7 | 69.4 | 53.6 | 73.0 | N/A | N/A | N/A |
| **SSR (Ours)** | 130M | 7B | 67.0 (↑2.3) | 82.1 (↑12.6) | 76.1 (↑22.5) | 73.3 (↑0.3) | N/A | N/A | N/A |

We illustrate the process to construct SSRBench as shown in Appendix E.2. First, we define 6 distinct task categories. Then, we randomly sampled image-text pairs from SSR-CoT, proportionally retaining the distribution of its original data sources. These samples were independently classified into task categories by GPT-4o [92] and Gemini-2.5-Pro [94]. Only instances for which both models agreed on the assigned category were included in SSRBench; instances with disagreement were returned to SSR-CoT.

In recent years, LLMs have demonstrated significant advancements in language understanding, reasoning, and text generation, exhibiting strong perceptual and comprehension capabilities through the implicit world knowledge they encapsulate. Therefore, LLMs have increasingly been used as assessors to evaluate generation performance in question-answering tasks [7, 1, 95, 96]. Consistent with our approach in data quality evaluation, we employ the Qwen2.5-14B-Instruct-1M [38, 93], a powerful LLM, to evaluate the performance of VLMs in this benchmark.

## 4.3 Implementation Details

In this paper, we utilize Mamba [71] as the lower-level efficient language model for reasoning, Qwen2.5 [38] as the LLM for alignment in the first training stage, and Qwen2.5-VL [91] as the VLM supporting multi-modal comprehension in the second training stage. During Stage 1, we exclusively train the MIDI component on our proposed SSR-CoT dataset. In Stage 2, we jointly train the SSR using both the SSR-CoT dataset and the LLaVA-Instruct-150K dataset [1]. Leveraging the efficiency of LoRA [97] and Fully Sharded Data Parallel (FSDP) [98], training SSR requires approximately 19 hours for Stage 1 and 48 hours for Stage 2, using a single Nvidia 8-H800 GPU node equipped with 80GB VRAM. Detailed hyperparameter configurations are provided in Appendix A.

## 4.4 Main Results

Table 3 presents the comparative performance of the SSR against its backbone and state-of-the-art baselines on SpatialBench [17] and SSRBench. As shown by the results, our SSR in 3 billion parameters can achieve comparable or even higher results than large-scale baseline models, including closed-source and backbone models. Our larger variant, comprising 7 billion parameters, yields the best performance on most tasks across the two benchmarks. Compared to the top-performing baselines in each benchmark, SSR exhibited notable improvements in the average question answering accuracy, achieving a maximum enhancement of 13.6 and an average improvement of 6.77. Moreover, we also provide a detailed analysis of the performance improvements compared to the backbone model across additional benchmarks in Section 5.1.

Table 5: Performance comparison on general VQA benchmarks

| Method | Size | | VQAv2 [104] | TextVQA [105] | POPE [106] | MMBench [107] | GQA [57] |
| | Mamba | VLM | | | | | |
|---|---|---|---|---|---|---|---|
| Qwen2.5-VL [91] | N/A | 3B | 72.5 | 57.0 | 84.4 | 75.9 | 56.2 |
| **SSR (Ours)** | 130M | 3B | 79.0 (↑6.5) | 61.3 (↑4.3) | 86.0 (↑1.6) | 78.3 (↑2.4) | 63.6 (↑7.4) |

Table 6: Performance comparison among the backbone model, the SSR with/without the second training stage. $_{PAP}$ indicates that the MIDI module was employed in a plug-and-play manner.

| Method | Size | | SpatialBench [17] | | | | SSRBench$^G$ | | | SSRBench$^S$ | | |
| | Mamba | VLM | Position | Existence | Counting | Size | Existence | Attribute | Action | Counting | Position | Object |
|---|---|---|---|---|---|---|---|---|---|---|---|---|
| Qwen2.5-VL [91] | N/A | 3B | 55.9 | 80.0 | 76.4 | 25.0 | 66.4 | 58.9 | 63.1 | 34.1 | 60.5 | 51.9 |
| **SSR$_{PAP}$** | 130M | 3B | 64.7 (↑8.8) | 80.0 (0.0) | 79.6 (↑3.2) | 30.0 (↑5.0) | 70.1 (↑3.7) | 59.5 (↑0.6) | 63.7 (↑0.6) | 36.6 (↑2.5) | 61.1 (↑0.6) | 53.2 (↑1.3) |
| **SSR** | 130M | 3B | 64.7 (↑8.8) | 80.0 (0.0) | 82.9 (↑6.5) | 31.7 (↑6.7) | 83.2 (↑16.8) | 82.1 (↑23.2) | 72.6 (↑9.5) | 51.2 (↑17.1) | 83.3 (↑22.8) | 74.7 (↑22.8) |

## 5 Analysis

### 5.1 Performance Improvement

We present additional experimental results in Table 4, demonstrating the improved performance of SSR compared to the backbone model across the five benchmarks shown in Table 11 at varying model scales. Specifically, across the three benchmarks reporting average values, SSR models of different sizes demonstrated average improvements of **11.2** and **9.4** compared to the backbone model. The most significant improvements were observed in the space task of the benchmark, where the enhancements reached **20.9** and **22.5**, respectively. This result exceeds the improvements reported in Table 2, indicating that our SSR effectively reasons about information highly relevant to multi-modal VQA tasks without introducing significant additional noise. Furthermore, the training paradigm of SSR enhances performance not only on two evaluation datasets closely related to the training data but also on the out-of-domain CV-Bench [102], VSR [82], What's Up [103] and multiple general VQA benchmarks [57, 104–107] as shown in Table 5. These findings indicate that our training approach effectively further improves the generality and generalization capability of the SSR in addition to enhanced spatial understanding performance.

### 5.2 Ablation Studies

As reported in Table 6, these experiments illustrate the performance of the MIDI module when integrated in a plug-and-play manner without second training stage, leading to improved spatial understanding. Specifically, this approach achieves average performance gains of **4.4** and **1.6** on different benchmark datasets. On certain tasks, the plug-and-play approach achieves performance improvements of up to **8.8**, demonstrating the effectiveness of this usage. In addition, after the second stage training, the performance of the complete SSR model will be significantly improved on this basis, achieving average performance gains of **5.7** and **18.7** on different benchmark datasets. Moreover, we provide case studies in Appendix F.

### 5.3 Efficiency

We evaluate the Qwen2.5-VL [91] after fine-tuning on SSR-CoT dataset, overly protracted and convoluted textual intermediate reasoning chains not only increase the risk of erroneous conclusions but also impose prohibitive computational costs that undermine inference efficiency, which can better reflect the importance of latent reasoning method in CoT application. As illustrated in Table 7, the results demonstrate that, although SSR introduces a modest absolute latency per generated token, its latent-reasoning paradigm dramatically curtails the number of CoT tokens needed to reach a final response. Consequently, under the CoT-based evaluation framework, the overall end-to-end inference speed is substantially improved.

Table 7: Inference efficiency comparison on SpatialBench [17].

| Model | Size | SpatialBench [17] | Token Per Sample | Token Per Second | Inference Time Per Sample |
|---|---|---|---|---|---|
| Qwen2.5-VL [91] (w/ SFT on SSR-CoT) | 3B | 51.3 | 437.28 | 18.88 | 23.16s |
| **SSR (Ours)** | 3B | 64.8 | 2.62 | 8.18 | 0.32s |

Table 8: LLM-Assistant Evaluation Performance comparison on SSRBENCH, evaluated using different LLMs. (For each result, the left column presents the original Qwen-based evaluation score, while the right column reports the corresponding GPT judgment).

| Model | Size | Existence | Attribute | Action | Counting | Position | Object |
|---|---|---|---|---|---|---|---|
| Qwen2.5-VL [91] | 3B | 66.4 / 58.9 | 58.9 / 59.5 | 63.1 / 67.5 | 34.1 / 34.1 | 60.5 / 58.6 | 51.9 / 52.6 |
| **SSR (Ours)** | 3B | 83.2 / 81.3 | 82.1 / 79.8 | 72.6 / 73.3 | 51.2 / 51.2 | 83.3 / 80.8 | 74.7 / 74.7 |

### 5.4 LLM-Assistant Evaluation

To mitigate potential biases arising from employing models of the same family as judges, we re-assessed the SSRBENCH results with GPT-4o-mini [92], the outcomes are reported in Table 8. High inter-model agreement between the evaluation scores assigned by different LLM judges in the table suggests that simple answer-comparison tasks largely resist bias within model series.

### 5.5 Rationale Embedding

To analyze whether MIDI effectively captures depth information and conducts spatial reasoning guided by rationale, we visualize the cosine similarity between latent tokens, both with and without rationale. Figure 6 visualizes the cosine similarities between the latent tokens produced in two different paradigms: x-axis: latent tokens inserted inside the rationale, y-axis: latent tokens inserted immediately after the question and used to start the answer generation. Diagonal cells represent these two states of the same sample. High values on the diagonal indicate that the model has learned to map the rationale to the latent representation, confirming that it successfully distills the spatial knowledge embedded in the rationale. Low off-diagonal values indicate that the latent tokens remain sample-specific and do not collapse to a generic representation.

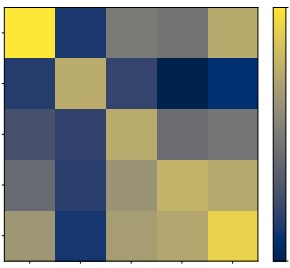

Figure 6: Cosine similarity matrix of reasoning latent tokens with/without rationale.

## 6 Conclusion

In this paper, we propose a novel VLM SSR with an important module named MIDI to interpret depth for enhancing the depth perception and spatial reasoning capabilities of existing VLMs. MIDI can even be efficiently integrated into existing VLMs in a seamless, plug-and-play manner. To enable effective training and evaluation, we curate a multi-modal CoT dataset SSR-COT and present a comprehensive benchmark SSRBENCH. Extensive experiments conducted across four distinct benchmarks demonstrate that SSR consistently achieves state-of-the-art performance enhancements over existing approaches, particularly excelling in spatially-oriented visual question answering tasks.

**Broader Impacts.** Our proposed SSR demonstrates that spatial reasoning capabilities can be incrementally enhanced without adversely affecting its existing VLM functionalities. This provides an innovative avenue for research communities to integrate additional capabilities into VLMs.

**Limitation and Future Works.** Although SSR shows astounding performance, this study is limited to the Qwen/Qwen-VL series; future work will broaden the VLM scope to test generalizability.

## Acknowledgments and Disclosure of Funding

This work was supported by the National Science and Technology Innovation 2030 - Major Project (Grant No. 2022ZD0208800), and NSFC General Program (Grant No. 62176215).

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

# A    Hyperparameters

Detailed hyperparameter configurations are provided in Table 9.

Table 9: Training hyper-parameters of our proposed SSR.

| Configuration | Stage 1 | Stage 2 | Configuration | Stage 1 | Stage 2 |
|---|---|---|---|---|---|
| Vision Encoder | Clip-ViT-Large-Patch14-336 [67] | | Optimizer | AdamW [108] | |
| Depth Encoder | Siglip-So400M-Patch14-384 [69, 109] | | Learning Rate | 0.00002 | |
| Mamba | Mamba 130M [71] | | Numerical Precision | BFloat16 | |
| LLM | Qwen2.5 3B [38] | N/A | Epoch | 2 | 1 |
| VLM | N/A | Qwen2.5-VL 3B [91] | Global Batch Size | 32 | 32 |
| Question Length | 256 | | Learning Schedule | Cosine Decay | |
| Rational Length | 1024 | N/A | Warm-up Ratio | 0.02 | |
| Answering Length | N/A | 256 | Number of Latent Tokens | 10 | |

# B    SSRBENCH Results in Score Metrics

The evaluation metrics employed for SSRBENCH include both accuracy and a quantitative score ranging from 0 to 5. Quantitative results are presented in Table 10, with detailed descriptions of the assessment methodology provided in Appendix D. These scores are generally consistent with the accuracy trends presented in Table 3.

Table 10: Score performance comparison on SSRBENCH. SSRBENCH $^{G}$ and SSRBENCH $^{S}$ denote general and spatial tasks, respectively.

| Method | Size | | SSRBENCH $^{G}$ | | | SSRBENCH $^{S}$ | | |
|---|---|---|---|---|---|---|---|---|
| | Mamba | VLM | Existence | Attribute | Action | Counting | Position | Object |
| PROPRIETARY | | | | | | | | |
| GPT-4o-mini [92] | N/A | N/A | 4.05 | 2.95 | 3.46 | 3.12 | 2.87 | 2.66 |
| Claude-3.5-Haiku [99] | N/A | N/A | 3.48 | 2.99 | 2.71 | 2.75 | 2.56 | 2.31 |
| OPEN-SOURCE | | | | | | | | |
| SpatialVLM [10, 100] | N/A | 3B | 2.34 | 3.32 | 3.55 | 2.34 | 3.56 | 3.24 |
| LLaVA-1.5 [2] | N/A | 7B | 4.17 | 3.72 | 3.66 | 2.71 | 3.87 | 3.56 |
| LLaVA-NeXT [86] | N/A | 7B | 4.23 | 3.59 | 3.79 | 2.66 | 3.69 | 3.41 |
| LLaVA-NeXT [86] | N/A | 13B | 4.30 | 3.82 | 3.79 | 2.76 | 3.78 | 3.12 |
| SpatialBot [17] | N/A | 3B | 3.97 | 3.47 | 3.82 | 2.66 | 3.96 | 3.47 |
| Emu3 [101] | N/A | 8B | 3.07 | 2.35 | 2.39 | 1.71 | 3.04 | 2.28 |
| Qwen2.5-VL [91] | N/A | 3B | 3.56 | 3.42 | 3.56 | 2.41 | 3.43 | 3.00 |
| Qwen2.5-VL [91] | N/A | 7B | 4.07 | 3.55 | 3.71 | 2.85 | 3.50 | 3.16 |
| **SSR (Ours)** | 130M | 3B | 4.44 | 4.28 | 3.95 | 3.17 | 4.40 | 4.02 |
| **SSR (Ours)** | 130M | 7B | 4.65 | 4.17 | 4.10 | 3.71 | 4.43 | 4.16 |

**Discussion with Meteor.**    Meteor [21] is an approach similar to ours, designed to compress rationales using efficient large language models. However, unlike our method, Meteor does not separate the reasoning module from the large language model during response generation. Due to this tight coupling, Meteor must be trained end-to-end from scratch, a process that demands extensive datasets and significant computational resources. In contrast, our method specifically focuses on enhancing the spatial awareness and reasoning abilities of Vision-Language Models (VLMs), leveraging their inherent capabilities to a greater extent. Consequently, our approach substantially reduces the complexity and resource requirements related to training VLMs from scratch. Moreover, we focus on directionally enhancing the depth perception and spatial reasoning capabilities of existing VLMs in this paper. Therefore, our comparative analysis primarily emphasizes evaluating model performance before and after applying these enhancements.

# C    SSR-CoT

As detailed in Section 4, the SSR-CoT is constructed from four distinct data sources, with spatially-aware CoT rationales generated for each data sample. Representative samples are shown in Figure 4.

Additionally, a comprehensive description of the rationale-generation pipeline specific to each data source is presented in Appendix C.2, C.1, and C.3.

## C.1 Visual-CoT

For Visual-CoT [52], each data sample includes a bounding box that serves as a CoT rationale to guide the generation of the corresponding answer. We utilize this bounding box, which is closely related to the target answer, as an intermediate step to query spatial information about the selected object using SpatialRGPT [65], a spatial question-answering model tailored for vertical domains. Subsequently, we aggregate the obtained spatial question-and-answer information using a powerful LLM such as GPT-4o [92]. The resulting text serves as the CoT rationale for the Visual-CoT data source within SSR-CoT.

---

**Spatial Query for Visual-CoT**

1. What is the object in [bbox]? Think step by step, and avoid repetition.
2. Can you estimate the height and width of [bbox]? Think step by step, and avoid repetition.
3. What is the object to the left of [bbox], and what is its height and width? Think step by step, and avoid repetition.
4. What is the object to the left of [bbox], and what is its distance to [bbox]? Think step by step, and avoid repetition.
5. What is the object to the right of [bbox], and what is its height and width? Think step by step, and avoid repetition.
6. What is the object to the right of [bbox], and what is its distance to [bbox]? Think step by step, and avoid repetition.
7. What is the object in front of [bbox], and what is its height and width? Think step by step, and avoid repetition.
8. What is the object in front of [bbox], and what is its distance to [bbox]? Think step by step, and avoid repetition.
9. What is the object behind [bbox], and what is its height and width? Think step by step, and avoid repetition.
10. What is the object behind [bbox], and what is its distance to [bbox]? Think step by step, and avoid repetition.
11. What is the object below [bbox], and what is its height and width? Think step by step, and avoid repetition.
12. What is the object below [bbox], and what is its distance to [bbox]? Think step by step, and avoid repetition.
13. What is the object above [bbox], and what is its height and width? Think step by step, and avoid repetition.
14. What is the object above [bbox], and what is its distance to [bbox]? Think step by step, and avoid repetition.

---

**Rationale Generation for Visual-CoT**

Please generate an image description in continuous paragraphs using these strict guidelines:

Coordinate Usage Rules:
1. ONLY use coordinates that are explicitly defined in this mapping table:
   - Region [0]: [bbox]
   - ...
2. Do NOT create or infer any new coordinates
3. Each coordinate can only be used ONCE in the description

4. Coordinates must be written in [x1,y1,x2,y2] format without spaces

Content Rules:
1. Place coordinate immediately after describing its corresponding object
2. Integrate coordinates naturally within complete sentences
3. Include all provided measurements and spatial relationships
4. Maintain narrative flow while incorporating technical details
5. Focus on visual elements and their relationships
6. Embed coordinates from the mapping table immediately after their corresponding region objects (e.g., "a dog [x1,y1,x2,y2]")
7. Maintain paragraph continuity by integrating coordinates within complete sentences
8. Preserve strict region-coordinate mapping from the provided table
9. Use only [x1,y1,x2,y2] format without spaces
10. Exclude technical metadata and region index numbers from final text
11. Automatically resolve spatial contradictions using coordinate data
12. Ensure coordinate annotations flow naturally after object nouns

Input Data:
Spatial Query and Response for Visual-CoT

## C.2 VoCoT

VoCoT [53] includes multiple bounding boxes per data sample, more than Visual-CoT [52], to clearly outline reasoning paths involving multiple objects within an image. Similar to the process used for Visual-CoT, we perform spatial queries on each object associated with a bounding box. Additionally, we capture the relative spatial relationships between every pair of objects to comprehensively utilize available spatial context and support accurate reasoning. Finally, we aggregate this spatially derived question-and-answer information using a robust language model, such as GPT-4o [92].

---

**Spatial Query for VoCoT**

***Query for each bounding box:***

1. What is the object in [bbox]? Think step by step, and avoid repetition.

2. Can you estimate the height and width of [bbox]? Think step by step, and avoid repetition.

3. What is the object to the left of [bbox], and what is its height and width? Think step by step, and avoid repetition.

4. What is the object to the left of [bbox], and what is its distance to [bbox]? Think step by step, and avoid repetition.

5. What is the object to the right of [bbox], and what is its height and width? Think step by step, and avoid repetition.

6. What is the object to the right of [bbox], and what is its distance to [bbox]? Think step by step, and avoid repetition.

7. What is the object in front of [bbox], and what is its height and width? Think step by step, and avoid repetition.

8. What is the object in front of [bbox], and what is its distance to [bbox]? Think step by step, and avoid repetition.

9. What is the object behind [bbox], and what is its height and width? Think step by step, and avoid repetition.

10. What is the object behind [bbox], and what is its distance to [bbox]? Think step by step, and avoid repetition.

11. What is the object below [bbox], and what is its height and width? Think step by step, and avoid repetition.

---

12. What is the object below [bbox], and what is its distance to [bbox]? Think step by step, and avoid repetition.

13. What is the object above [bbox], and what is its height and width? Think step by step, and avoid repetition.

14. What is the object above [bbox], and what is its distance to [bbox]? Think step by step, and avoid repetition.

*Query for every two bounding box:*

1. Which one is higher between [bbox1] and [bbox2]? Think step by step, and avoid repetition.

2. Can you estimate how far apart [bbox1] and [bbox2] are? Think step by step, and avoid repetition.

3. What direction is [bbox2] in relation to [bbox1]? Think step by step, and avoid repetition.

4. How far is [bbox1] from [bbox2] horizontally? Think step by step, and avoid repetition.

5. Does [bbox1] have a larger size compared to [bbox2]? Think step by step, and avoid repetition.

6. Does [bbox1] have a lesser width compared to [bbox2]? Think step by step, and avoid repetition.

---

**Rationale Generation for VoCoT**

Integrate all measurements values and spatial information from the conversation into answer to get detailed reasoning rationale with spatial details.
Then, extract the direct question and answer from question and answer respectively.

Content Rules:
1. Place coordinate immediately after describing its corresponding object first time, make sure each coordinate appear only once.
2. Avoid introducing other coordinates that do not appear in answer.
3. Add all provided measurements values and spatial relationships from the conversation to the rationale detailedly.
4. Ensure the rationale contains all the information from each sentence in the conversation, especially the measurements values and spatial relationships.
5. Automatically resolve spatial contradictions using coordinate data based on the image.

Output in the following json template:
{
    "question": <question>
    , "rationale": <rationale>
    , "answer": <answer>
}

Question: Question
Answer: Answer
Conversation: Spatial Query and Response for VoCoT

---

## C.3 SpatialQA

Unlike Visual-CoT [52] and VoCoT [53], the SpatialQA [17] dataset does not provide intermediate CoT reasoning steps or bounding boxes for object identification. Therefore, we leverage GPT-4o [92], a powerful multi-modal large language model, to generate detailed synthetic rationale data. These

synthetic rationales supply the necessary intermediate reasoning processes to enable accurate answer generation.

---

**Rationale Generation for SpatialQA**

I have an image and a question that I want you to answer.
I need you to strictly follow the format with four specific sections: summary, caption, reasoning, and conclusion.
It is crucial that you adhere to this structure exactly as outlined and that the final answer in the conclusion matches the standard correct answer precisely.
To explain further:
   - In summary, briefly explain what steps you'll take to solve the problem.
   - In caption, describe the contents of the image, specifically focusing on details relevant to the question.
   - In reasoning, outline a step-by-step thought process you would use to solve the problem based on the image.
   - In conclusion, give the final answer in a direct format, and it must match the correct answer exactly. If it's a multiple choice question, the conclusion should only include the option without repeating what the option is.
Finally, integrate these sections into a natural thinking paragraph.

Here's the question and answer:
Question: Question
Answer: Answer

---

## D   LLM-Assistant Evaluation

As discussed in Section 4, we utilize the LLM-Assistant evaluation method to assess the data quality of SSR-CoT and measure the performance of SSRBENCH [7, 1, 95, 96]. Evaluation metrics include accuracy and a quantitative score ranging from 0 to 5; both metrics are determined by the LLM-Assistant powered by the Qwen2.5-14B-Instruct-1M [38, 93].

---

**Prompt for LLM-Assistant VQA Evaluation**

You are an intelligent chatbot designed for evaluating the correctness of generative outputs for question-answer pairs.
Your task is to compare the predicted answer with the correct answer and determine if they match meaningfully. Here's how you can accomplish the task:
——
##INSTRUCTIONS:
- Focus on the meaningful match between the predicted answer and the correct answer.
- Consider synonyms or paraphrases as valid matches.
- Evaluate the correctness of the prediction compared to the answer.
Please evaluate the following image-based question-answer pair:
Question: question
Correct Answer: answer
Predicted Answer: response
Provide your evaluation only as a yes/no and score where the score is an integer value between 0 and 5, with 5 indicating the highest meaningful match.
Please generate the response in the form of a Python dictionary string with keys 'pred' and 'score', where value of 'pred' is a string of 'yes' or 'no' and value of 'score' is in INTEGER, not STRING.
DO NOT PROVIDE ANY OTHER OUTPUT TEXT OR EXPLANATION. Only provide the Python dictionary string.
For example, your response should look like this: {'pred': 'yes', 'score': 4.8}.

---

# E Benchmarks

## E.1 Benchmark Employed

We evaluate our method using various benchmarks: SpatialBench [17], CV-Bench [102], VSR [], What's Up [], our proposed SSRBENCH, and multiple general VQA benchmarks []. Table 11 summarizes the statistics for the all benchmarks. Tables 3 and 6 present comparisons between SSR and baseline methods, along with comprehensive ablation studies conducted on SpatialBench and SSRBENCH. Furthermore, Table 4 summarizes the performance improvements observed across all spatial-related benchmarks.

Table 11: Statistics of benchmarks utilized in this paper.

| Benchmark | Task | | Size | Benchmark | Task | | Size |
|---|---|---|---|---|---|---|---|
| SpatialBench [17] | Position | | 34 | CV-Bench [102] | Count | | 788 |
| | Existence | | 40 | | Relation | | 650 |
| | Counting | | 20 | | Depth | | 600 |
| | Size | | 40 | | Distance | | 600 |
| | | TOTAL | *114* | | | TOTAL | *2638* |
| SSRBENCH | General | Existence | 107 | | Spatial | Counting | 41 |
| | | Attribute | 168 | | | Position | 162 |
| | | Action | 157 | | | Object | 154 |
| | | | | | | TOTAL | *789* |
| VSR [82] | Random | | 1222 | | Zero-Shot | | 2195 |
| | | | | | | TOTAL | *3417* |
| What's Up [103] | | TOTAL | *820* | VQAv2 [104] | | TOTAL | *107k* |
| TextVQA [105] | | TOTAL | *5k* | POPE [106] | | TOTAL | *9k* |
| MMBench [107] | | TOTAL | *3k* | GQA [57] | | TOTAL | *12k* |

## E.2 SSRBENCH

To construct the SSRBENCH dataset, we first filter data samples from SSR-CoT. Subsequently, we feed each filtered sample into the multi-modal large language models GPT-4o [92] and Gemini-2.5-Pro [94], using the following prompt to classify the task category. As mentioned in Section 4 and shown in Figure 7, if the classification results from both models are consistent, the sample is added to SSRBENCH; otherwise, it is returned to SSR-CoT.

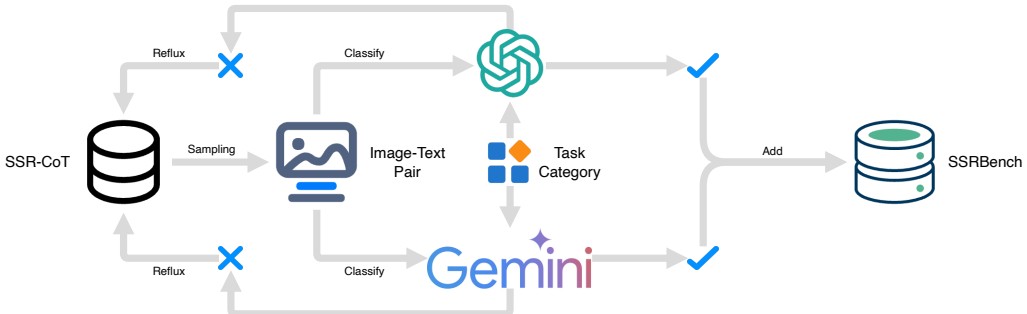

Figure 7: Schematic of SSRBENCH construction pipeline.

---

**Task classification for SSRBENCH**

You are an expert in image-based question classification.
You need to classify each input question into a specific task type based on the following

---

taxonomy.
Task Categories:
 Spatial:
  Explanation: Involve identifying and understanding the position, size, shape, and relative relationships of objects in an image.
   Subtasks:
    Count: Counting objects in the image (e.g., questions like "How many ...?").
    Relative Position Recognition: Determining spatial relations like "to the left of", "above", or "on the right".
    Position Based Object Recognition: Identifying an object based on its spatial relation to another object (e.g., "What is the object to the left of the dog?").
 General:
  Explanation: Involve classifying, recognizing, or reasoning about visual content without necessarily focusing on spatial relations.
   Subtasks:
    Existence: Determining whether an object or feature is present (e.g., "Is there a cat?").
    Attribute Recognition: Identifying attributes like color, texture, size, or state (e.g., "What color is the apple?").
    Action Recognition: Recognizing what action or activity is occurring (e.g., "What is the man doing?").

For each input question:
 First determine whether the question belongs to the spatial or general category.
 Then classify it into one of the three subtasks under that category.
 If the question does not match any of the subtasks under either category, return None.

Output format:
 {"category": "spatial" or "general", "subtask": "subtask_name" or "None"}

Example Input: "Is there a bicycle in the image?"
Example Output: {"category": "general", "subtask": "existence"}

Now, let's begin classification. Here's the question:
Question: Question

# F   Case Studies

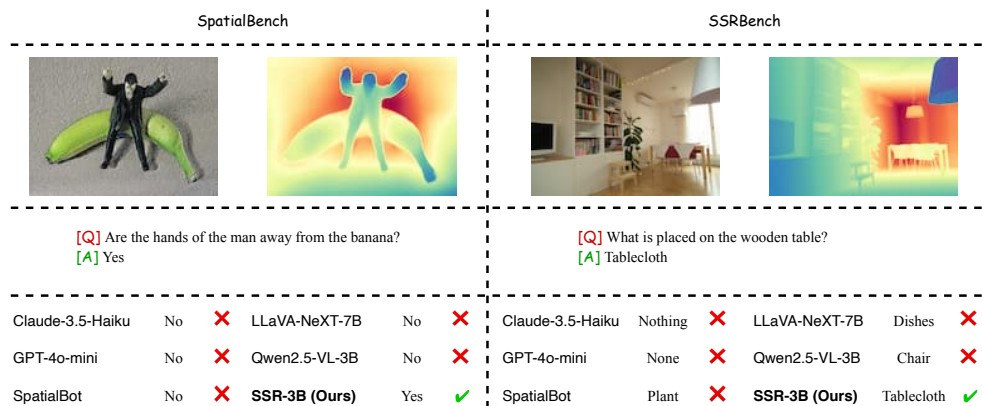

Figure 8: Two examples illustrating question-answering performance by baseline models compared to our SSR are presented.

To further illustrate the effectiveness of our proposed SSR, we provide two example cases in Figure 8, comparing the performance of SSR against five baseline models: Claude-3.5-Haiku [99], GPT-4o-mini [92], SpatialBot [17], LLaVA-NeXT-7B [86], and the backbone model Qwen2.5-VL-3B [91].

As shown, our SSR consistently produces correct answers, whereas all baseline models fail to provide accurate responses.

In the left example, the images depict only people and bananas. Consequently, the model must abandon conventional assumptions and carefully reason about the spatial relations explicitly present in the image to answer accurately. In the right example, complex relationships among numerous objects are depicted, and relevant features for answering the posed question are not immediately obvious. In this case, the model must thoroughly comprehend the correspondence between each object and the given question, as well as understand intricate spatial relations among these objects, to produce a correct response. These examples clearly demonstrate that our SSR effectively enhances the spatial awareness and reasoning capabilities of vision-language models, thereby significantly improving their ability to understand complex spatial relationships.

