# OpenReview forum: "SSR: Enhancing Depth Perception in Vision-Language Models via Rationale-Guided Spatial Reasoning"
_NeurIPS.cc/2025/Conference — NeurIPS 2025 poster_

### Official Review · Reviewer_aoD6 · 2025-06-25

**Clarity:** 3
**Significance:** 3
**Originality:** 3
**Rating:** 5
**Confidence:** 4

**Summary:**

This paper introduces SSR (Spatial Semantic Reasoning), a novel paradigm that redefines depth integration in vision-language models (VLMs) through interpretable and structured rationale-language representations. Furthermore, the proposed framework transforms raw depth data into latent embeddings via knowledge distillation, thereby reducing computational costs. SSR provides a plug-and-play method to improve the spatial reasoning capabilities of existing VLMs. Finally, for a comprehensive evaluation, the author constructs a large-scale vision-language dataset (SSR-COT) and a comprehensive multi-task benchmark (SSR-BENCH).

**Questions:**

If the authors can address all of my major concerns, I will raise my rating; otherwise, I will lower the current rating. Please correct me if I've misunderstood any aspects of the work.

**Ethical Concerns:**

["NO or VERY MINOR ethics concerns only"]

**Final Justification:**

Although the authors have addressed my major concerns, some minor weaknesses remain. I hope the authors can conduct additional experiments to provide a more reliable estimate of variance in the final version, which would also further enhance the overall quality of the paper. Therefore, I will maintain my original positive rating.

**Limitations:**

yes

**Quality:**

3

**Strengths And Weaknesses:**

Strengths:

1. SSR demonstrates superior comprehension of depth maps and effectively enhances the spatial reasoning capabilities of the VLM.

2. The collection, annotation, and quality evaluation pipeline for the large-scale vision-language dataset (SSR-CoT) are well-designed, providing high-quality spatial Chain-of-Thought (CoT) reasoning data.

3. The SSRBench benchmark thoroughly assesses VLM spatial reasoning performance.

Major Weaknesses:

Q1. My primary concern regards the training-free SSR_{PAP} module, which demonstrates only marginal performance gains on certain tasks, particularly on the proposed SSRBench. While this limited improvement is understandable given that pretrained VLMs have never seen similar depth latent features before, could the authors explain why the SSR method can operate training-free and be directly plugged into a pretrained VLM model?

Q2. Following Q1, if SSR undergoes Stage 2 training, would this lead to pretrained knowledge forgetting or a degragation in inherent reasoning capabilities? For example, how would SSR-7B perform on other general- or common-sense reasoning benchmarks compared to its backbone, Qwen2.5-VL?

Q3. For the method part, if SSR does not convert depth information into latent features but directly generates language representations as additional text input for the VLM, how would this affect spatial reasoning performance?

Minor Weaknesses:

Q4. In Table 3, while the 7B backbone (Qwen2.5-VL) outperforms the 3B version on all metrics, why does SSR-3B surpass SSR-7B on certain metrics? Is this due to random variability, or does the effectiveness of the proposed method (SSR) diminish as model parameters scale up?

Q5. In Figure 1, the right-side VQA example contains an error: the question is multiple-choice, yet the model responds "No."

---

> ### Author Rebuttal · Authors · 2025-07-30
>
> Thanks for your careful and valuable comments. We will explain your concerns point by point.
>
> **Q1**: The latent tokens were aligned into semantic space of LLM through training stage 1, thus VLM based on LLM as backbone can understand the meaning of token to a certain extent.
>
> **Q2**: Limited by time and computational resource constraints, we provide supplementary experiments for the following general VQA tasks based on the SSR-3B model:
>
> | Model      | Size | VQAv2     | TextVQA   | POPE      | MMBench   | GQA       |
> | ---------- | ---- | --------- | --------- | --------- | --------- | --------- |
> | Qwen2.5-VL | 3B   | 72.54     | 57.01     | 84.44     | 75.85     | 56.18     |
> | SSR        | 3B   | **79.04** | **61.33** | **86.00** | **78.26** | **63.58** |
>
> Our proposed SSR model outperform Qwen2.5-VL on multiple general benchmarks, this demonstrates that the pre-training knowledge of the Qwen2.5-VL will not be forgotten, but more common knowledge will be learned during training.
>
> **Q3**: We evaluate the Qwen2.5-VL after fine-tuning on SSR-CoT dataset, overly protracted and convoluted textual intermediate reasoning chains not only increase the risk of erroneous conclusions but also impose prohibitive computational costs that undermine inference efficiency:
>
> | Model                          | Size | SpatialBench | SSRBench$^{\text{G}}$ | SSRBench$^{\text{S}}$ | CV-Bench  |
> | ------------------------------ | ---- | ------------ | -------------- | -------------- | --------- |
> | Qwen2.5-VL (w/ SFT on SSR-CoT) | 3B   | 51.28        | 38.40          | 28.60          | 56.73     |
> | SSR                            | 3B   | **59.52**    | **79.30**      | **69.73**      | **68.90** |
>
> | Model                          | Size | SpatialBench | Token Per Sample | Token Per Second | Inference Time Per Sample |
> | ------------------------------ | ---- | ------------ | ---------------- | ---------------- | ------------------------- |
> | Qwen2.5-VL (w/ SFT on SSR-CoT) | 3B   | 51.28        | 437.28           | **18.88**        | 23.16                     |
> | SSR                            | 3B   | **59.52**    | **2.62**         | 8.18             | **0.32**                  |
>
> **Q4**: We thank the reviewer for this perceptive observation. We attribute SSR-3B's occasional superiority over SSR-7B on SpatialBench metrics primarily to two factors: (1) training both models on the identical dataset may inadequately support the larger model's capacity, and (2) SpatialBench's limited size increases susceptibility to statistical fluctuations. Finally, owing to computational constraints we could not perform multiple random restarts to obtain a reliable variance estimate. These factors jointly confound any firm conclusion about a occasional drop in SSR’s effectiveness as parameters scale. We will explicitly discuss these limitations in the final version and investigate them more thoroughly in future work.
>
> **Q5**: The erroneous example depicted in Figure 1 will be amended, with the corrected version also provided in Figure 5.

---

> > ### Comment · Reviewer_aoD6 · 2025-08-05
> >
> > Thank you for the authors’ rebuttal response and effort. The authors have addressed my major weaknesses from Q1 to Q3. Specifically, the improved performance on general- or common-sense reasoning benchmarks empirically demonstrates that the proposed method not only enhances spatial-related reasoning but also preserves the original reasoning capabilities. This is an important validation of the paper’s effectiveness.
> >
> > Regarding my minor weaknesses from Q4 and Q5, I think the authors have not fully addressed my Q4 concern. I hope they can conduct more experiments to obtain a reliable variance estimate in the final version.
> >
> > Overall, I believe the paper presents a practical plug-and-play method for improving the spatial reasoning capabilities of VLMs, so I will maintain my accept rating. Since other reviewers have not yet responded, I will continue to follow the discussion and feedback.

---

> > > ### Author Response · Authors · 2025-08-06
> > >
> > > Sincerely thank you for your response. Glad to see that our additional experiments addressed your questions and concerns. We will conduct more experiments to obtain a reliable variance estimate and report the results in final version. Your thorough review and constructive feedback on our paper greatly enhanced the quality and completeness of our work, we will revise the paper accordingly.

---

### Official Review · Reviewer_JmB8 · 2025-06-26

**Clarity:** 2
**Significance:** 3
**Originality:** 3
**Rating:** 4
**Confidence:** 4

**Summary:**

This paper focuses on enhancing and evaluating spatial understanding leveraging depth images and reasoning traces or rationales. First of all, authors present two datasets, SSR-CoT and SSR-Bench, which are built semi-automatically from existing datasets, such as LLaVA-CoT, Visual-CoT, VO-CoT and SpatialQA. Both datasets are VQA datasets, where rationales are added together with the answers. SSR-CoT is built for training purposes, and SSR-Bench is directly extracted from SSR-CoT. As the second contribution, authors present a new framework to build VLMs with spatial understanding. The main idea is to leverage RGB and depth images together with rationales to train a new module called MIDI. MIDI is designed to produce latent tokens that aggregate the information from RGB, depth and answer rationales, which can be used as the input for a standard VLM for general VQA tasks. Authors evaluate their approach on various datasets, including their SSR-Bench showing improvements.

**Questions:**

1. In line 20, authors state: "relying solely on RGB is inadequate for accurately capturing spatial information such as relative positions and distances". I do not agree with this statement. Indeed, the depth information used by your system comes from RGB images, which means that in principle, that information could be represented by VLMs in their latent representations. Do you have any evidence to support this claim?
2. From line 37 to line 43: I understand this text is written to motivate your work. However, I think it is too vague. You should better specify what VLMs are missing for spatial understanding. By the way, what does this sentence mean? "existing methods incorporate depth explicitly without capitalizing on its inferential value". Could you clarify it please?
3. From line 47 to 53: similar to the previous. I think the text is too vague. You should better motivate the usage of language-based rationales.
4. Typos: lines 77 and 78: there are two "has" that sould be "have".
5. Line 113: Please, rewrite the sentence.
6. Line 155: "an efficient language within the MIDI" -> "an efficient language model within the MIDI"?
7. Equations 1 and 2 are the same loss function, aren't they? Why do you use two different names to refer to them? "Standard causal modeling loss" and "standard cross-entropy loss for auto-regressive generation". Did I miss anything here?
8. Line 200: authors mention that they perform sampling quality checks. Can you explain how you do so?
9. Line 245: detailed hyperparameters in Appendix A, but how did you set those hyperparameters? I couldn't find any information about the procedure.
10. Table 4: I don't know MME in detail, but it's very weird to me to see the employed performance metric (MME reports summation). It is very difficult to understand the differences in performance between different systems. Could you clarify this point better, please?
11. Table 5: It is not clear to me what PAP means here. I think you refer to the usage of latent tokens generated by MIDI as the result of the Stage 1 training, directly as the input of VLMs, but without any Stage 2 training. Is that right? In that case, please make it more clear in the text.
12. Figure 6: I don't understand this figure and its motivation. Legends for the axes are missing and the text in Section 5.3 is not very descriptive. It may be my fault, but I read it three times and I don't understand what this is showing us.

**Ethical Concerns:**

["NO or VERY MINOR ethics concerns only"]

**Final Justification:**

I raised my score to 4, because the authors show that their approach actually improves over strong baselines. The evaluation methodology is now clear and correct, and the experiments performed are significant. Besides, authors also contribute an interesting dataset for multimodal spatial reasoning. However, I still have concerns about the clarity and writing of the paper. The authors promised to improve it, but even in their rebuttal text they have many typos. That is why I score the paper with a 4.

**Limitations:**

Yes

**Quality:**

3

**Strengths And Weaknesses:**

**Strengths:**
1. The SSR-CoT and SSR-Bench datasets are interesting.
2. Leveraging depth images estimated from monocular RGB images is also an interesting idea.
3. The plug-and-play nature of the latent tokens learnt by MIDI, which could be used for any VLM (although this is not proven in the experimentation).

**Weaknesses:**
1. The writing of the paper can be improved in my opinion. The paper lacks clarity at many points and makes some claims that are very arguable in my opinion (I list some of those points in the Questions section).
2. Many of the design decisions are not justified. Examples: i) the selection of vision encoders for RGB and depth images (why CLIP for one and SigLIP for the other?), ii) the usage of Mamba for the MIDI module (why not a standard transformer-based LM?), iii) why do they use an LLM to align MIDI latent tokens semantically and not the VLM itself (which has an LLM that could be used and seems a more natural option), iv) why do they use Qwen2.5 as the alignment LLM and not another LLM? I'm not saying the design decisions are wrong, but I think those decisions should be justified in the paper.
3. The experimental methodology has many flaws from my point of view.
- I think some baselines are missing to support the main claims of the paper, which are, to the best of my understanding, that depth images and language-based rationales are key to improve spatial understanding of VLMs. One of the baselines is Qwen2-5-VL (their base VLM) directly fine-tuned in SSR-CoT, without the MIDI component, but with the rationales provided by the dataset. Another important baseline could be their whole system but without the depth images. I think those baselines are needed to really understand the contribution of depth images and rationales in the results.
- The selection of evaluation datasets is not justified and is arguable. I think that some spatial understanding datasets are missing, such as Visual Spatial Reasoning [1] and What's Up [2]. On the other hand, I don't understand why MME is used, when the dataset is not designed for spatial understanding (and I have no evidence spatial understanding is key for improved results; I may have missed something here).
- The evaluation protocol is weak. They use a judge LLM, but they do not provide any human correlation information. They blindly believe the output of the judge, which is not a good practice. On the other hand, the selection of the judge LLM is not suitable. In general, LLMs tend to evaluate better their own outputs, and as far as I know, it is not recommended to use the same LLM family to evaluate competing LLMs. In this case, their system is based on Qwen2.5-VL, whose backbone LLM is Qwen-2.5, and the judge is also a Qwen2.5 LLM (the 14B variant).

[1] Liu, Fangyu, Guy Emerson, and Nigel Collier. "Visual spatial reasoning." Transactions of the Association for Computational Linguistics 11 (2023): 635-651.

[2] Kamath, Amita, Jack Hessel, and Kai-Wei Chang. "What’s “up” with vision-language models? Investigating their struggle with spatial reasoning." In Proceedings of the 2023 Conference on Empirical Methods in Natural Language Processing, pp. 9161-9175. 2023.

---

> ### Author Rebuttal · Authors · 2025-07-30
>
> Thanks for your careful and valuable comments. We will explain your concerns point by point.
>
> **W1**: We will revise the paper per your suggestions and carefully review all expressions in the paper.
>
> **W2 i)**: Consistent with the prevailing paradigm in modern VLMs (e.g., LLaVA), we adopt CLIP as the RGB encoder. For depth modalities, we follow the SpatialBot design and employ SigLIP.
>
> **W2 ii)**: As shown in Table 3(a) of the original Meteor paper, the Mamba model has inherent computational efficiency based on linear complexity and strong long sequence modeling capability. With the same parameter count, BERT, XLNet, and GPT all require roughly twice the computation of Mamba, highlighting its clear advantage in throughput.
>
> **W2 iii)**: We align the latent tokens with the LLM rather than with the feature space of a particular VLM, because our objective is to situate the tokens within a universal semantic space, like CLIP, thereby maximizing cross-model generalizability. Consequently, after Stage 1 training, MIDI can be seamlessly transferred to diverse VLMs in a plug-and-play fashion:
>
> | Model                           | Size | SpatialBench | SSRBench$^{\text{G}}$ | SSRBench$^{\text{S}}$ | CV-Bench |
> | ------------------------------- | ---- | ------------ | -------------- | -------------- | -------- |
> | InternVL3                       | 8B   | 66.15        | 68.90          | 58.90          | 75.43    |
> | MIDI + InternVL3$_{\text{PAP}}$ | 8B   | **72.73**    | **75.97**      | **71.73**      | **83.8** |
>
> As illustrated in the table below, when MIDI is combined with InternVL3-8B without any task-specific adaptation, it achieves markedly superior performance, underscoring both the generalization capacity and the broad applicability of our approach.
>
> **W2 iv)**: The majority of experiments in this paper employ models from the Qwen series, which have emerged as one of the most widely adopted open-source large language models in the research community. Their extensive usage, like InternVL series, offers a well-established and readily available foundation, thereby facilitating the seamless integration and evaluation of our proposed approach.
>
> **W3 a)**: Thanks for your suggestions, we supplement these contrastive experiments to provide a clearer understanding of the role of depth information:
>
> | Model                          | Size | SpatialBench | SSRBench$^{\text{G}}$ | SSRBench$^{\text{S}}$ |
> | ------------------------------ | ---- | ------------ | -------------- | -------------- |
> | Qwen2.5-VL (w/ SFT on SSR-CoT) | 3B   | 51.28        | 38.40          | 28.60          |
> | SSR$_{\text{PAP}}$ (w/o Depth) | 3B   | 55.98        | 63.50          | 49.07          |
> | SSR$_{\text{PAP}}$             | 3B   | 57.20        | 64.37          | 50.97          |
>
> We also provide the performance indicators of the qwen2.5-vl base model after adding depth information. Please refer to the response for reviewer 5EXr's W3, where the analysis description will also be provided.
>
> **W3 b)**: Both Visual Spatial Reasoning and What’s Up were constructed prior to the era of LLM; owing to time and resource constraints, we report zero-shot performance instead of task-specific fine-tuning on these datasets in the table below:
>
> | Model      | Size | VSR$_{\text{Random}}$ | VSR$_{\text{Zero-Shot}}$ | What's Up |
> | ---------- | ---- | --------------------- | ------------------------ | --------- |
> | Qwen2.5-VL | 3B   | 72.98                 | 76.43                    | 85.37     |
> | SSR        | 3B   | **78.63**             | **82.90**                | **87.93** |
>
> As for MME benchmark, our primary objective is to demonstrate that the proposed method directionally strengthens the VLM’s spatial-reasoning capabilities without degrading its performance on general vision–language tasks, please refer to the supplementary experiments we provided to Reviewer aoD6's Q2 for more results.
>
> **W3 c)**: As mentioned in lines 231–236, the LLM-Assistant evaluation only requires a straightforward textual comparison between the model’s response and the ground-truth answer, which is a easy operation for contemporary, highly-capable LLMs. To mitigate potential biases arising from employing models of the same family as judges, we re-assessed the SSRBench results with `GPT-4o-mini`, the outcomes are reported in the table below (For each result, the left column presents the original Qwen-based evaluation score, while the right column reports the corresponding GPT judgment):
>
> | Model      | Size | Existence   | Attribute   | Action      | Counting    | Position    | Object      |
> | ---------- | ---- | ----------- | ----------- | ----------- | ----------- | ----------- | ----------- |
> | Qwen2.5-VL | 3B   | 66.4 / 58.9 | 58.9 / 59.5 | 63.1 / 67.5 | 34.1 / 34.1 | 60.5 / 58.6 | 51.9 / 52.6 |
> | SSR        | 3B   | 83.2 / 81.3 | 82.1 / 79.8 | 72.6 / 73.3 | 51.2 / 51.2 | 83.3 / 80.8 | 74.7 / 74.7 |
>
> The close agreement between the evaluation scores assigned by different LLM judges in the table indicates that simple answer-comparison tasks are largely immune to bias arising from the use of models within the same series.
>
> **Q1**: RGB images alone provide insufficient spatial information for robust spatial understanding, as evidenced by recent studies (e.g., SpatialVLM, SpatialBot, SpatialRGPT). To address this limitation, existing work focuses on enriching VLMs with explicit spatial cues. While depth information is typically derived from RGB data, it offers a more effective spatial representation by distilling critical distance relationships. Conversely, raw RGB data may introduce ancillary visual noise that obscures spatial reasoning. This principle parallels the role of optical flow in video understanding field: though computationally extracted from RGB sequences, optical flow provides essential motion semantics that significantly enhances neural network performance, which demonstrating how processed geometric representations can surpass raw pixel data in conveying structural information.
>
> **Q2**: We consider that what existing VLMs lack is the ability to perceive spatial information and reasoning. Spatial perception indicates identify the position relationship, real distance between different objects and so on, which are ambiguous to obtain from rgb image. The reasoning ability denotes the integrate and extract the portion of this spatial information that is beneficial to answering the question. The sentence means input depth information directly into VLMs without any adaptation like SpatialBot, we argue that this compromises the efficient use of depth information. We will correct the misquote of this sentence in the final version, thank you for your careful reading.
>
> **Q3**: Thanks for your insightful suggestion. In the final version, we will expand the corresponding paragraph to elucidate why and how SSR converts raw depth maps into a structured latent representation within in semantic space. In addition, we will explicitly highlight the SSR’s superior generalization capacity and computational efficiency supported by supplementary experiments (Response to W2 iii and Response to Reviewer KarK’s Q2).
>
> **Q4 - Q6**: We sincerely appreciate your meticulous suggestions; all typos will be corrected in the final version.
>
> **Q7**: The two equations are indeed the same mathematical loss (causal, next-token cross-entropy), but they are applied to different model inputs and targets, and this difference is the key point. Equation 1 is computed on the reasoning chain.  Input: $Z_{R}, Y_{R,<i}$, Target: $Y_{R,i}$, Goal: train the lightweight LLM to (i) perform implicit reasoning correctly and (ii) keep its latent representations aligned with the frozen LLM’s semantic space. Equation 2 is computed on the final answer tokens. Input: $Y_{A,<j}$, Target: $Y_{A,j}$, Goal: train the VLM to interpret the latent tokens emitted by the LLM and generate the correct textual answer. Thus, the loss are identical in form but operate on disjoint data streams and serve different training objectives in the two-stage pipeline.
>
> **Q8**: Similar to the quality-assessment protocol shown in Table 2, we randomly draw 10 % of the cached samples and evaluate VQA accuracy both with and without their generated rationales. Rationales that improve accuracy are retained and incorporated into the final SSR-CoT dataset; those that degrade performance are discarded, and the all samples in cache are re-submitted for re-annotation.
>
> **Q9**: Owing to the high computational costs inherent in training LLMs, hyper-parameters were configured based on a comprehensive synthesis of prior empirical experience and existing papers.
>
> **Q10**: The MME score shown in Table 4 is indeed an aggregate (sum) of multiple sub-task accuracies. To verify that the targeted spatial-enhancement does not harm general capability, we evaluate the SSR model on more general visual-language tasks, please see our detailed response to Reviewer aoD6's Q2.
>
> **Q11**: PAP indicates the MIDI module is directly applied to the pre-trained VLM in a plug-and-play manner after the training stage 1, without co-training. We will add a clearer explanation to the final version.
>
> **Q12**: Figure 6 visualizes the cosine similarities between the latent tokens produced in two different paradigms: x-axis: latent tokens inserted inside the rationale, y-axis: latent tokens inserted immediately after the question and used to start the answer generation. Diagonal cells represent these two states of the same sample. High values on the diagonal indicate that the model has learned to map the rationale to the latent representation, confirming that it successfully distills the spatial knowledge embedded in the rationale. Low off-diagonal values indicate that the latent tokens remain sample-specific and do not collapse to a generic representation. For improved clarity in the final version, axis labels and enhanced descriptions will be added.

---

> > ### Comment · Reviewer_JmB8 · 2025-08-01
> >
> > Thank you for all the clarifications and new results. I am mostly satisfied with the authors' responses. However, I have some specific comments:
> >
> > **W3:** The role of the MME dataset should be explained in the paper. In the answer to Reviewer aoD6, you also show the results for other generic VL datasets. Given those results, I would recommend the authors to keep all those datasets together and show them in the paper with the suitable explanations.
> >
> > **W3:** Regarding JudgeLLM-based evaluation, I am now OK. I would recommend authors to add both judges to their results to avoid any doubts.
> >
> > **Q7:** Understood. In that case, use the same name for both losses.
> >
> > **Q8:** Explain this in the paper, since I believe it is important.
> >
> > **Q9:** Add those explanations to the paper.
> >
> > **Q10:** In the response to Reviewer aoD6 you don't use the same metric for MME as in the paper. If there are several sub-tasks, and accuracy is used for all of them, why don't you provide the average of those accuracies?
> >
> > I am willing to raise my score of the paper given the answer of the authors, but I have to admit that the clarity of the paper is still a concern for me.

---

> ### Author Response · Authors · 2025-08-02
>
> **Q10**: we report all tasks separately following LLaVA [1] to provide a comprehensive understanding of the performance metrics for Reviewer aoD6. We will present the average values of these tasks in Table 4 for enhanced readability, while detailed metrics for each individual task will be provided in the appendix for thorough analysis.
>
> We sincerely appreciate your thorough review and constructive feedback, which has significantly enhanced the quality and completeness of our work. We will incorporate all your valuable suggestions into our paper in an appropriate manner.
>
> *[1] Haotian Liu, Chunyuan Li, Yuheng Li and Yong Jae Lee. "Improved Baselines with Visual Instruction Tuning". In Proc. of CVPR, pp. 26286–26296, 2024.*

---

### Official Review · Reviewer_5EXr · 2025-07-02

**Clarity:** 4
**Significance:** 3
**Originality:** 3
**Rating:** 4
**Confidence:** 4

**Summary:**

This paper aims to address a common limitation in existing vision-language models (VLMs), where they rely primarily on RGB images and thus lack accurate spatial understanding capabilities. To alleviate this limitation, the paper introduces the new Spatial Sense and Reasoning (SSR) approach, which leverages depth information to generate structured and interpretable rationales in text. Additionally, the approach also uses knowledge distillation to compress the abovementioned rationales into latent embeddings. To train the resulting model, the authors curate the new SSR-CoT dataset as well as introduce SSRBENCH which is a benchmark that evaluates on multiple downstream tasks.

**Questions:**

Please look at the abovementioned weaknesses. It will be much more insightful to include empirical comparisons to more VLMs that are targeted at spatial reasoning and more ablations.

**Ethical Concerns:**

["NO or VERY MINOR ethics concerns only"]

**Final Justification:**

The authors have addressed my concerns.

**Limitations:**

Yes

**Quality:**

3

**Strengths And Weaknesses:**

Strength 1 - In terms of clarity and quality of the paper, the paper is relatively well-written and motivated. The model figures are informative and especially helpful in helping the reader to understand the proposed SELongVLM approach.

Strength 2 - With regards to the significance of the work, this paper introduces an interesting method to integrate depth information, which is well-aligned with the latent language embedding space of the base LLM. This can be especially useful and practical since the introduced Mamba-based Image-Depth Interpreter (MIDI) modoule can be integrated into existing VLMS without having to retrain them. This helps to save time and cost for adapting such VLMs to new tasks or improving their performance directly on benchmarks.

Strength 3 - This work also introduces a new large-scale training dataset SSR-CoT and the SSRBENCH evaluation benchmark. The former can be useful for the research community to integrate into their existing data curation and training pipelines for training VLMs that are more capable of spatial reasoning.

Weakness 1 - The proposed SSR approach somewhat lacks novelty. While integrating depth information for spatial reasoning in VLMs effectively is an important research problem, there are already numerous recent works such as SpatialVLM, SpatialBot and SpatialRGPT that have explored adding depth and spatial information to VLMs. It is unclear how different SSR is from the abovementioned three works.

Weakness 2 - More importantly and related to Weakness 1, the empirical results included in the paper do not compare to some VLMS that are specifically targeted at spatial reasoning, such as SpatialVLM and SpatialRGPT among many others. It will be much more insightful if such empirical comparisons are included.

Weakness 3 - Finally, while the reported performance of SSR across multiple benchmarks appear to be strong, it is difficult to understand the relative importance of the different components proposed in the paper.  For instance, the paper does not include any detailed ablation on whether using depth information alone could provide substantial performance gains without generating the intermediate chain-of-thoughts rationale.

---

> ### Author Rebuttal · Authors · 2025-07-30
>
> Thanks for your careful and valuable comments. We will explain your concerns point by point.
>
> **W1**: A key differentiator between our proposed SSR framework and existing approaches (SpatialVLM, SpatialBot, SpatialRGPT) lies in the portability of spatial understanding capabilities. Specifically, our MIDI module enables zero-shot transfer of spatial reasoning abilities to diverse VLMs without requiring retraining. This is achieved by translating spatial information into latent tokens within the semantic space, allowing seamless integration. Notably, as demonstrated in our response to Reviewer JmB8's W2 iii, when integrated with latest powerful VLMs like InternVL3, the MIDI module enables performance exceeding Qwen2.5-VL. In contrast, adapting the existing methods to newer, more capable VLMs necessitates complete re-adaptation and retraining, which is a process demanding significant computational resources.
>
> **W2**: Thank you for suggesting these spatial understanding baselines. SpatialVLM has not released official code or checkpoints; we therefore evaluated the community-recommended third-party re-implementation version:
>
> | Model                  | Size | SpatialBench | SSRBench$^{\text{G}}$ | SSRBench$^{\text{S}}$ | CV-Bench  |
> | ---------------------- | ---- | ------------ | -------------- | -------------- | --------- |
> | SpatialVLM (3rd Party) | 3B   | 57.66        | 51.23          | 50.97          | 61.53     |
> | SSR                    | 3B   | **59.52**    | **79.30**      | **69.73**      | **68.90** |
>
> We will incorporate this results into Table 3 in the final version. Regarding SpatialRGPT, we note that it is an object-centric approach that expects one or more target objects to be supplied with bounding boxes or segmentation masks in every query. Designing a protocol that fairly aligns this requirement with our setting is non-trivial, and we were unable to complete such an adaptation within the current submission deadline. We will explore appropriate evaluation methods and update the evaluation results in time.
>
> **W3**: Thanks for your suggestions, to comprehensively evaluate our SSR method, we conducted supplementary ablation studies focused on depth information. Specifically, we established two baselines: (1) directly integrating depth data into the backbone model, and (2) removing depth information from the plug-and-play setting. We then rigorously compared the performance of SSR against these baselines:
>
> | Model                          | Size | SpatialBench | SSRBench$^{\text{G}}$ | SSRBench$^{\text{S}}$ |
> | ------------------------------ | ---- | ------------ | -------------- | -------------- |
> | Qwen2.5-VL                     | 3B   | 53.80        | 62.80          | 48.83          |
> | Qwen2.5-VL (RGB + Depth)       | 3B   | 52.46        | 62.57          | 49.03          |
> | SSR$_{\text{PAP}}$ (w/o Depth) | 3B   | 55.98        | 63.50          | 49.07          |
> | SSR$_{\text{PAP}}$             | 3B   | 57.20        | 64.37          | 50.97          |
>
> As shown in above table, directly integrating depth information into the backbone model impairs its understanding capability, leading to performance degradation. However, spatial-related CoT learning systematically enhances the model's spatial reasoning capacity. Subsequent incorporation of depth information further amplifies this spatial understanding, demonstrating the complementary effect of depth data when processed through the CoT framework.

---

> > ### Comment · Reviewer_5EXr · 2025-08-06
> >
> > I would like to thank the authors for their comprehensive efforts in responding to my questions as well as those raised by other reviewers. After reading the authors' responses and other comments, I think the responses have addressed my concerns.

---

> > > ### Author Response · Authors · 2025-08-07
> > >
> > > Sincerely thank you for your response. Glad to see that our response addressed your questions and concerns. Your thorough review and constructive feedback on our paper greatly enhanced the quality and completeness of our work. We are willing to address any concerns preventing score improvement in the discussion phase.

---

### Official Review · Reviewer_karK · 2025-07-03

**Clarity:** 3
**Significance:** 3
**Originality:** 4
**Rating:** 4
**Confidence:** 3

**Summary:**

To enhance spatial reasoning, SSR translates raw depth data into structured textual rationales as intermediate representations. These rationales are distilled into latent embeddings via the MIDI module and then passed as tokens into a vision-language model (VLM) to answer spatial reasoning questions. Experiments show that SSR significantly improves spatial reasoning performance across multiple tasks while preserving general VLM capabilities.

In addition, the authors introduce SSR-COT, a million-scale vision-language dataset with spatial annotations, and SSRBENCH, a comprehensive benchmark for evaluating spatial understanding. Together, these resources support robust evaluation and training of spatial reasoning models.

**Questions:**

1. Could the authors provide qualitative examples or a detailed analysis of common failure cases to better understand when and why the method underperforms and outperforms baseline models?

2. How does the proposed MIDI module affect efficiency? Can the authors report the computational or memory overhead compared to a standard vision-language model during training and inference?

**Ethical Concerns:**

["NO or VERY MINOR ethics concerns only"]

**Final Justification:**

**Response to rebuttal**

I thank the authors for their effort and response. My concerns have been addressed. Overall, I believe the paper makes a significant contribution to improving the spatial reasoning capabilities of VLMs by incorporating depth information and chain-of-thought reasoning. I will keep my original positive score.

**Limitations:**

Limitations have been discussed.

**Paper Formatting Concerns:**

No format concerns.

**Quality:**

3

**Strengths And Weaknesses:**

Strengths
1. Novel depth integration approach: The transformation of raw depth information into interpretable textual rationales offers a more meaningful way to incorporate spatial cues compared to direct depth feature fusion, enabling better spatial reasoning.

2. Comprehensive evaluation framework: The introduction of both a large-scale dataset (SSR-COT with over 1 million samples) and the SSRBENCH benchmark provides valuable resources for the community to evaluate spatial reasoning capabilities.

Weaknesses

1. Limited analysis of failure modes: The paper lacks a qualitative discussion of when and why the proposed approach fails, or how it improves upon baseline models in specific scenarios.

2. Unclear computational overhead: While the paper introduces the MIDI module, it does not provide a clear analysis of inference-time computational cost or memory usage compared to a vanilla vision-language model.

---

> ### Author Rebuttal · Authors · 2025-07-30
>
> Thanks for your careful and valuable comments. We will explain your concerns point by point.
>
> **W1 & Q1**: Due to conference policy, we are unable to provide full image cases. However, we have analyzed error cases and can describe key phenomena verbally. For *counting* tasks, most failures stem from ambiguous, unclear, or boundary-indistinct target objects (e.g., two overlapping giraffes being misidentified as one), indicating the requirement for more fine-grained, object-oriented visual understanding. Similarly, failure in other tasks arise from issues like small, unclear objects (e.g., a child’s shoe in the image being misrecognized as a sandal due to its pattern), highlighting the need for VLMs to receive and comprehend higher-resolution images.
>
> **W2 & Q2**: We added inference time and memory usage statistics when supplementing general task experiments for Reviewer aoD6's Q2. Across all evaluated tasks, the computational consumption remains highly consistent; a representative result for POPE benchmark is summarised in the table below:
>
> | Model      | Size | Accuracy  | Token Per Second | Memory Usage |
> | ---------- | ---- | --------- | ---------------- | ------------ |
> | Qwen2.5-VL | 3B   | 56.18     | **12.08**        | **20.46**    |
> | SSR        | 3B   | **63.58** | 8.29             | 34.32        |
>
> Moreover, we counted the relevant resource consumption data when fine-tuning Qwen2.5-VL with SSR-CoT for Reviewer JmB8's W3, which can better reflect the importance of Latent Reasoning method in CoT application:
>
> | Model                          | Size | SpatialBench | Token Per Sample | Token Per Second | Inference Time Per Sample |
> | ------------------------------ | ---- | ------------ | ---------------- | ---------------- | ------------------------- |
> | Qwen2.5-VL (w/ SFT on SSR-CoT) | 3B   | 51.28        | 437.28           | **18.88**        | 23.16s                    |
> | SSR                            | 3B   | **59.52**    | **2.62**         | 8.18             | **0.32s**                 |
>
> The results demonstrate that, although SSR introduces a modest absolute latency per generated token, its latent-reasoning paradigm dramatically curtails the number of CoT tokens needed to reach a final response. Consequently, under the CoT-based evaluation framework, the overall end-to-end inference speed is substantially improved.
>
> We appreciate your careful review and insightful suggestions. Accordingly, the aforementioned experimental results will be incorporated into Section 5.

---

> > ### Comment · Reviewer_karK · 2025-08-06
> > **Response to the authors.**
> >
> > I thank the authors for their effort and response. My concerns have been addressed. Overall, I believe the paper makes a significant contribution to improving the spatial reasoning capabilities of VLMs by incorporating depth information and chain-of-thought reasoning. I will keep my original positive score.

---

> > > ### Author Response · Authors · 2025-08-06
> > >
> > > Sincerely thank you for your positive response. It is great to see that our additional supplementary experiments and the responses during the rebuttal period have completely addressed your questions and concerns. Your thorough review and constructive feedback on our paper greatly enhanced the quality and completeness of our work, we will revise the paper accordingly.

---

### Decision · Program_Chairs · 2025-09-17

**Decision:**

Accept (poster)

**Comment:**

This paper was reviewed by four experts who universally recommended the paper for acceptance.  This paper presents an approach that incorporates depth information into a VLM to boost spatial reasoning performance.  While this has been studied in prior work, results in the rebuttal demonstrate that the proposed approach better utilizes this information by aligning the depth information with text tokens rather than directly using the depth information.  While there are some areas of concern that prevent a higher recommendation, such as missing error bars or a more extensive comparison to closely related work that was raised in the rebuttal, the paper provides enough evidence for their approach that it merits acceptance.  The authors are strongly encouraged to consider reviewer comments for clarity and requests for empirical evidence that can further bolster their contributions for their camera ready.